# Interaction Between Glucagon-like Peptide 1 and Its Analogs with Amyloid-β Peptide Affects Its Fibrillation and Cytotoxicity

**DOI:** 10.3390/ijms26094095

**Published:** 2025-04-25

**Authors:** Ekaterina A. Litus, Marina P. Shevelyova, Alisa A. Vologzhannikova, Evgenia I. Deryusheva, Alina V. Chaplygina, Victoria A. Rastrygina, Andrey V. Machulin, Valeria D. Alikova, Aliya A. Nazipova, Maria E. Permyakova, Victor V. Dotsenko, Sergei E. Permyakov, Ekaterina L. Nemashkalova

**Affiliations:** 1Institute for Biological Instrumentation, Pushchino Scientific Center for Biological Research of the Russian Academy of Sciences, 142290 Pushchino, Russia; marina.shevelyova@gmail.com (M.P.S.); lisiks.av@gmail.com (A.A.V.); janed1986@ya.ru (E.I.D.); shadowhao@yandex.ru (A.V.C.); certusfides@gmail.com (V.A.R.); alikovalera@mail.ru (V.D.A.); alija-alex@rambler.ru (A.A.N.); mperm1977@gmail.com (M.E.P.); permyakov.s@gmail.com (S.E.P.); elnemashkalova@gmail.com (E.L.N.); 2Skryabin Institute of Biochemistry and Physiology of Microorganisms, Pushchino Scientific Center for Biological Research of the Russian Academy of Sciences, 142290 Pushchino, Russia; and.machul@gmail.com; 3Department of Organic Chemistry and Technologies, Kuban State University, 149 Stavropolskaya St., 350040 Krasnodar, Russia; victor_dotsenko_@mail.ru

**Keywords:** diabetes mellitus, glucagon-like peptide-1, liraglutide, exenatide, semaglutide, Alzheimer’s disease, amyloid-β peptide (Aβ), protein–protein interaction, Aβ fibrillation, Aβ cytotoxicity

## Abstract

Clinical data as well as animal and cell studies indicate that certain antidiabetic drugs, including glucagon-like peptide 1 receptor agonists (GLP-1RAs), exert therapeutic effects in Alzheimer’s disease (AD) by modulating amyloid-β peptide (Aβ) metabolism. Meanwhile, the direct interactions between GLP-1RAs and Aβ and their functional consequences remain unexplored. In this study, the interactions between monomeric Aβ40/Aβ42 of GLP-1(7-37) and its several analogs (semaglutide (Sema), liraglutide (Lira), exenatide (Exen)) were studied using biolayer interferometry and surface plasmon resonance spectroscopy. The quaternary structure of GLP-1RAs was investigated using dynamic light scattering. The effects of GLP-1RAs on Aβ fibrillation were assessed using the thioflavin T assay and electron microscopy. The impact of GLP-1RAs on Aβ cytotoxicity was evaluated via the MTT assay. Monomeric Aβ40 and Aβ42 directly bind to GLP-1(7-37), Sema, Lira, and Exen, with the highest affinity for Lira (the lowest estimates of equilibrium dissociation constants were 42–60 nM). GLP-1RAs are prone to oligomerization, which may affect their binding to Aβ. GLP-1(7-37) and Exen inhibit Aβ40 fibrillation, whereas Sema promotes it. GLP-1 analogs decrease Aβ cytotoxicity toward SH-SY5Y cells, while GLP-1(7-37) enhances Aβ40 cytotoxicity without affecting the cytotoxic effect of Aβ42. Overall, GLP-1RAs interact with Aβ and differentially modulate its fibrillation and cytotoxicity, suggesting the need for further studies of our observed effects in vivo.

## 1. Introduction

Alzheimer’s disease (AD) is a neurodegenerative disease characterized by a gradual decline in cognitive abilities and memory impairment, which significantly complicate social and professional activities. Worldwide, approximately 416 million individuals are affected by AD dementia, prodromal AD, or preclinical AD, accounting for 22% of the population aged 50 years and older [1]. To date, the U.S. Food and Drug Administration (FDA) has approved nine drugs for the treatment of AD, of which only three (aducanumab, lecanemab, and donanemab) are used for pathogenetic therapy and target the reduction of amyloid-β peptide (Aβ) deposits [2,3]. At the same time, the use of these drugs is associated with side effects, such as brain edema and microhemorrhages [4]. Aβ plays a central role in AD pathology [5]. It is derived from a transmembrane amyloid precursor protein (APP) through sequential proteolytic cleavage by β-secretase and γ-secretase [6]. The predominant forms of Aβ are peptides comprising 38, 40, or 42 residues, named Aβ38, Aβ40, and Aβ42, respectively [7,8]. Monomeric Aβ is intrinsically disordered and therefore prone to aggregation, with the formation of short fibrillar oligomers, most cytotoxic globular nonfibrillar oligomers, and mature amyloid fibrils [9].

Epidemiological data indicate a strong link between AD and diabetes mellitus (DM); patients with diabetes have a 65% increased risk of developing AD [10,11]. DM is a severe chronic disease that has a serious impact on the life and well-being of individuals, families, and society. Globally, at least 529 million people suffer from diabetes [12]. Type 2 diabetes mellitus (DM2) is the most prevalent form of DM (90% of all cases), characterized by high blood glucose levels (hyperglycemia) and insulin resistance [13]. The last one, along with neuroinflammation, oxidative stress, increased levels of advanced glycosylation end products, mitochondrial dysfunction, metabolic syndrome, and the accumulation of Aβ and tau protein in the brain, are common features of AD and DM2 (reviewed in [14]). Therefore, type 3 diabetes, which manifests as insulin resistance in brain tissue, affecting cognitive function and contributing to AD progression, has recently been proposed as a brain-specific type of DM [15].

Clinical studies in patients with mild cognitive impairment and AD have demonstrated that administration of certain antidiabetic medications, including intranasal insulin, metformin, incretins, and thiazolidinediones, can improve cognition and memory (reviewed in [16]). Incretins (glucagon-like peptide 1 (GLP-1) and gastric inhibitory peptide) are gut hormones that are secreted after nutrient intake and act on pancreatic β-cells to enhance glucose-stimulated insulin secretion [17]. Incretin-based therapy (including truncated GLP-1 and its derivatives) is playing an increasingly important role in the treatment of DM2 due to its efficacy and safety [18,19].

The *N*-terminally truncated forms of GLP-1, GLP-1(7-36)/(7-37), secreted from intestinal L cells [20], control meal-related glycemic excursions by augmentation of insulin expression and secretion and inhibition of glucagon release (reviewed in [21]). Some population of neurons in the nucleus tractus solitarii of the brainstem can also express GLP-1 [22,23]. GLP-1 can cross the blood–brain barrier (BBB) [24] and acts through the GLP-1 receptor, GLP-1R, which is expressed in several brain regions, including the hypothalamus, cerebral cortex, amygdala, hippocampus, caudate putamen, and globus pallidum [25]. GLP-1 signaling is important for cognition, and preclinical studies evidence the neuroprotective action of GLP-1 [26,27]. Murine GLP-1R contributes to control of synaptic plasticity and memory formation [28]. GLP-1R-deficient mice have a learning-deficient phenotype that can be rescued through hippocampal GLP1R gene transfer, while rats overexpressing GLP-1R in the hippocampus show improved memory and learning abilities [29]. GLP-1(7-36) has been shown to reduce Aβ levels in the mouse brain in vivo and to decrease levels of APP in cultured neuronal cells [30]. Similarly, GLP-1(7-36) protects cultured hippocampal neurons against Aβ/iron-induced death [30], while mutated GLP-1 rescues SH-SY5Y cells from Aβ42-induced apoptosis [31].

GLP-1 is efficiently inactivated by dipeptidyl peptidase-4 (DPP-4) and neutral endopeptidase 24.11, resulting in a plasma half-life of GLP-1 of approximately 1.5–5 min [32,33,34]. To overcome this limitation, DPP-4 inhibitors and long-acting GLP-1R agonists (GLP-1RAs) resistant to proteolysis by DPP-4 have been developed for clinical use (reviewed in [32,35]), including exenatide (Exen), liraglutide (Lira), semaglutide (Sema), etc.

Exen (trade name Byetta) consists of 39 amino acid residues with 53% homology to human GLP-1(7-37) (Figure 1A,C) and is resistant to DPP-4-mediated inactivation [36]. Exen decreases Aβ toxicity and oxidative stress in primary neuronal cultures and SH-SY5Y cells, interferes with the development of cognitive impairment, and significantly reduces brain levels of APP and Aβ in animal AD models [37,38,39,40]. Reduced Aβ accumulation in response to Exen has been shown in both mouse and worm AD models [40,41]. Evaluation of Exen’s effect in patients with moderate Parkinson’s disease showed sustained improvements in cognitive and motor measures [42]. Meanwhile, a pilot study of Exen in AD did not reveal significant differences in clinical, cognitive, or biomarker outcomes compared with the placebo, except for a reduction in Aβ42 levels in extracellular vesicles [43].

Another long-acting GLP-1 derivative, Lira (brand names Victoza and Saxenda), differs from GLP-1(7-37) by K34R substitution and palmitic acid attached to the K26 residue through a glutamic acid spacer (Figure 1A,E). The attached fatty acid chain favors binding to serum albumin, thereby slowing the clearance of Lira [44,45]. Lira has been shown to alleviate neuronal insulin resistance and to reduce Aβ formation and tau hyperphosphorylation in SH-SY5Y cells [46]. Tests of Lira in APP/PS1 AD mice show that it crosses the BBB, prevents memory loss and hippocampal deterioration, increases the number of young neurons in the dentate gyrus, and reduces neuronal inflammation, Aβ oligomer and APP levels, and Aβ plaque formation [47,48,49]. In clinical trials involving AD patients, Lira was found to prevent the decline of brain glucose metabolism; however, it did not significantly affect Aβ accumulation or cognition [50]. Functional magnetic resonance imaging revealed significant improvement in intrinsic connectivity in the default mode network in the group of persons at risk for AD taking Lira, but without detectable cognitive differences between the study groups [51].

Sema (trade names Ozempic, Wegovy, etc.) is a prolonged-release form of Lira with increased affinity for HSA, suitable for once-weekly administration [52]. Compared to Lira, Sema contains 2-aminoisobutyric acid at position 2 (prevents breakdown by DPP-4) and differs in structure of the fatty acid chain (C18 di-acid chain) and its linker (Figure 1A,B). Sema protects SH-SY5Y cells from Aβ25–35 by enhancing autophagy and inhibiting apoptosis [53]. The neuroprotective and anti-inflammatory properties of Sema were shown in a rat model of stroke [54]. Recent studies using human AD brain organoids have shown that Sema decreases levels of Aβ and phosphorylated tau levels. Additionally, in APP/PS1 transgenic mice, Sema improves cognitive performance, particularly learning and memory, and reduces amyloid plaque [55]. The oral form of Sema is currently being tested in patients with early AD in phase 3 clinical trials (NCT04777396 and NCT04777409) [56,57].

Despite encouraging clinical data and animal and cellular studies on the role of GLP-1 and its analogs in AD progression, information on their direct interaction with Aβ is lacking. To fill this gap, in the present study we probe the interaction between several GLP-1RAs and monomeric Aβ40/42 using biolayer interferometry (BLI) and surface plasmon resonance (SPR) spectroscopy. Furthermore, we assess the effects of GLP-1RAs on Aβ fibrillation and Aβ cytotoxicity toward SH-SY5Y cells.

## 2. Results

### 2.1. Interaction Between GLP-1RAs and Monomeric Aβ

Aβ40/Aβ42 was immobilized of the surface of the BLI sensor by amine coupling using EDAC/sulfo-NHS, followed by removal of the non-covalently bound Aβ molecules with 0.5% SDS solution, which ensured the monomeric state of Aβ. Passage over the sensor of 4–50 µM solutions of GLP-1(7-37), Exen, Lira, and Sema in buffer, simulating the conditions of the extracellular space, resulted in concentration-dependent sensograms characteristic of association/dissociation phases (Figure 2). Some of the resulting kinetic curves were successfully fitted using either a single binding site model (1) or a heterogeneous ligand scheme (2) (Figure 2). The resulting parameters of the GLP-1RA-Aβ interactions are summarized in Table 1. Meanwhile, some of the kinetic data were not consistent with these interaction models. Nevertheless, the clear signs of these interactions evidenced that their equilibrium dissociation constants, *K_D_*, reached the level of the analyte concentrations used in the BLI experiments, i.e., 25–50 µM for GLP-1(7-37) and 4–15 µM for Exen. The highest affinity for Aβ40/Aβ42 was observed for Lira, with *K_D_* values of 42–60 nM at protein concentrations of 5–10 µM (Table 1). Sema was 2.4–2.6 orders of magnitude less specific to Aβ40/Aβ42 (*K_D_* values of 11–22 µM) at protein concentrations of 17–38 µM.

The analogous examination of Aβ40/Aβ42 affinity for GLP-1RAs using SPR spectroscopy and Aβ as a ligand (Figure 3) yielded 1–1.5 orders of magnitude higher lowest *K_D_* estimates for Sema and Lira (Table 2), which may have been due to differences in the buffer conditions or analyte concentrations used in the BLI and SPR experiments. For Exen, the quality of the SPR data (Figure 3C,F) was insufficient for a reliable kinetic analysis, but it can be concluded that the corresponding *K_D_* values reached the analyte concentration level of 1 μM. The latter estimate was slightly lower than that derived from the BLI experiments, which could be rationalized by the same factors.

The *K_D_* estimates for Aβ-Sema/Lira complexes (Table 1 and Table 2) were comparable to those for Aβ binding to its natural depot, human serum albumin (HSA) (~0.1 μM [58]), as well as for Aβ complexes with fragments of the receptor for advanced glycation end products, which exhibit neuroprotective activity in both in vitro and in vivo models [59]. Similarly, the *K_D_* values for Aβ-Sema/Lira complexes were close to the *K_D_* estimate for the binding of ^125^I-labelled Lira to the GLP-1 receptor (1.3 × 10^−7^ M) [60]. Moreover, these values were close to the peak plasma concentrations of Sema/Lira (20–120 nM [61,62]), indicating that Sema/Lira interactions with Aβ (0.5 nM in plasma [63]) may occur in circulation.

### 2.2. Concentration-Dependent Changes in Quaternary Structure of GLP-1RAs

Since GLP-1RAs are prone to oligomerization and fibrillation [64,65,66], we studied the quaternary structure of GLP-1 and its analogs using dynamic light scattering (DLS) spectroscopy in a buffer with salt conditions close to physiological ones and similar to the BLI experiments (Table 3). A decrease in DLS sensitivity at protein concentrations of 0.05–0.02 mg/mL prevented measurement at GLP-1RA concentrations below 6–12 μM, depending on the peptide.

The main light scattering peak of 5–83 μM GLP-1(7-37) corresponded to particles with a hydrodynamic radius (*R_h_*) exceeding 92 nm, which indicated strong oligomerization of the peptide and explained the inability to describe analytically the BLI data on its interaction with Aβ40/Aβ42 (Figure 1A,E).

The DLS data for 6–105 μM Lira showed an increase in its degree of multimerization, *MW_Rh_*/*MW_m_*, with protein concentrations from 6.5–8.2 to 15.6, consistent with other reports [66,67,68]. This transition in the oligomeric state of Lira correlated with a tendency to change the *K_D_* values for its interaction with Aβ40/Aβ42 (Table 1).

Similarly to Lira, Exen demonstrated an increase in the degree of multimerization with protein concentrations (15–234 μM) from 1.6 to 5.9, in agreement with the literature data [69].

The *R_h_* estimates for Sema (12 μM, 47 μM) were consistent with its monomer and dimer, which were below the previous estimates for the formulation buffer composition [66]. Hence, the *K_D_* values for the interactions between Sema and Aβ40/Aβ42 estimated using BLI (Table 1) were close to the corresponding thermodynamic constants. By contrast, in the other cases complicated by the oligomerization of GLP-1RAs, the estimates shown in Table 1 represent only apparent constants.

Overall, the DLS data indicated that GLP-1Ras, at the concentrations used in the BLI experiments, existed as mixtures of oligomers with varying degrees of multimerization. However, since the degree of multimerization decreased with decreasing protein concentration, at plasma concentrations of 1–119 pM for GLP-1/Exen) [70,71,72] and 20–120 nM for Sema/Lira [61,62], GLP-1RAs predominantly existed in monomeric form. In this state, hydrophobic residues and fatty acid moieties were more accessible for interactions with Aβ, which likely facilitated binding.

### 2.3. Effect of GLP-1RAs on Aβ Fibrillation

The influence of GLP-1RAs on Aβ40 fibril formation at 30 °C was studied using the ThT fluorescence assay at ThT and GLP-1RA concentrations of 10 μM for both components (Figure 4). While Lira had no significant effect on fibrillation (Figure 4A), the other GLP-1RAs demonstrated drastically different behavior (Figure 4B). GLP-1(7-37) and Exen both suppressed Aβ40 fibrillation, whereas in the presence of Sema there was a clear tendency to stimulate the fibrillation process.

To explore the structural features of the grown fibrils, we examined them using negative-staining transmission electron microscopy, TEM (Figure 5). The Aβ40 sample and samples with the addition of Lira/Sema revealed dense clusters of intertwined mature fibrils up to 2 µm long (Figure 5A–C). When analyzing the samples with the addition of Exen, only scattered fibrils and shorter fibrils compared to the others were visible (Figure 5D). In addition, in the presence of Exen, large clusters of fibrils, which were characteristic of the other samples, did not form (Figure 5D). When analyzing the samples with the addition of GLP-1, sporadic small fibril clusters were still observed (Figure 5E,F). In summary, the microscopic data supported the findings from the ThT fluorescence assay.

Apparently, the ability of a particular GLP-1RA to affect Aβ fibrillation in vivo depends not only on its in vitro activity but also on its ability to penetrate the CNS and distribute across brain regions. Since Exen and GLP-1 readily cross the BBB [24,73], and GLP-1 can be expressed by some population of neurons [22,23], these GLP-1RAs have the potential to also suppress Aβ fibrillation in brain tissue. On the contrary, Sema does not cross the BBB [74], indicating that its ability to stimulate Aβ fibrillation in vitro (Figure 4B and Figure 5C) is unlikely to be of physiological significance.

Our in vitro data for Exen were consistent with the data showing reduced Aβ accumulation in AD models [40,41]. Interestingly, Lira and Sema are also able to reduce Aβ deposits in AD mouse models [47,48,49,55]. These GLP-1 analogs did not inhibit the process of fibrillation (Figure 4 and Figure 5), but in this case other mechanisms were probably involved, such as influence on APP level [47], on insulin signaling and insulin secretion level [75,76,77], as well as reduction of chronic inflammation level [12,49].

### 2.4. Structural Modeling of Complexes Between Aβ40 or Its Protofibril and GLP-1(7-37)/Exen

To identify structural patterns in the formation of GLP-1RA-Aβ complexes, the tertiary structures of the complexes between GLP-1(7-37)/Exen and the Aβ40 monomer were modeled using the ClusPro docking server [78] (Figure 6A,B). Additionally, we investigated the interaction between GLP-1(7-37)/Exen and the protofibrillar form of Aβ40, since this type of interaction (as well as interaction with monomeric Aβ40) may underlie the inhibitory effects of GLP-1(7-37)/Exen on Aβ40 fibril formation that we observed.

The modeling of the complex between GLP-1(7-37) and the Aβ40 monomer predicted (Figure 6A) that Aβ40 bound GLP-1(7-37) via *N*-terminal residues R5, D7, G9, Y10, and Q15 from the α-helix. The predicted Aβ40-binding site of GLP-1(7-37) included residues L20, E27, F28, W31, L32, K34, and G35. The modeling of structure of the Exen complex with the Aβ40 monomer predicted (Figure 6B) that Aβ40 bound Exen via residue L17 (α-helix) and C-terminal residues L34 and V39. The predicted Aβ40-binding site of Exen included *N*-terminal H1, residues F6, L10, Q13, M14, E17, and L21 (α-helix), and C-terminal residues P38 and S39. Thus, the predicted Aβ40-binding sites of GLP-1(7-37)/Exen were located in the region of residues 5–21 a.a. Meanwhile, the predicted contact residues of the Aβ40 molecule differed significantly for GLP-1(7-37) and Exen, which may reflect limitations of the rigid body approximation employed in the docking algorithm.

The modeling of the complex between GLP-1(7-37) and the Aβ40 protofibril predicted (Figure 6C) that GLP-1(7-37) interacted with chains A and C of the protofibril via residues T13, S14, S18, Y19, E21, Q23, A24, A25, E27, F28, W31, L32, K34, and G35. The chains A and C were predicted to bind GLP-1(7-37) via residues Q15, K16, V18, F19, F20, and V39. Note that the same residues of GLP-1(7-37) (L20, E27, F28, W31, L32, K34, and G35) participated in the binding of both the Aβ40 monomer and Aβ40 protofibril.

The analogous modeling of the complex between Exen and the Aβ40 protofibril predicted (Figure 6D) that Exen interacted with chains A and C of the protofibril via residues H1, G2, E3, G4, T5, F6, S8, L10, Q13, M14, E15, E17, R20, and S39. The chains A and C were predicted to bind Exen via residues Q15, V18, and V36. The residues H1, F6, L10, Q13, M14, and E17 were common for the binding sites of Exen with the Aβ40 monomer and Aβ40 protofibril.

The structural modeling results may explain the inhibition of Aβ40 fibrillation by GLP-1(7-37)/Exen observed in vitro (Figure 4 and Figure 5), since Aβ40 residues may be involved in binding to GLP-1(7-37)/Exen but in the formation of mature fibrils.

### 2.5. Effect of GLP-1RAs on Aβ Cytotoxicity Toward Human Neuroblastoma Cells

Since the deleterious effects of Aβ on neuronal cells are thought to be mediated by its oligomeric forms with increased cytotoxicity [9,79,80], we compared the cytotoxicity of Aβ40/Aβ42 alone and in the presence of GLP-1(7-37) or its analogs against human neuroblastoma SH-SY5Y cells using the MTT assay. GLP-1RAs were premixed with Aβ40/Aβ42 at an equimolar ratio in serum-free medium and added to the SH-SY5Y cells cultured in the same medium to a final concentration of both components of 10 µM. Staining with MTT was performed after incubation of the cells for 48 h.

In the absence of Aβ, Lira, Sema, and Exen had no effect on the survival of SH-SY5Y cells, whereas GLP-1(7-37) enhanced cell viability by 33% (Figure 7A). The addition of Aβ40 or Aβ42 alone decreased cell survival by 22% and 37%, respectively (Figure 7B,C). The addition of Lira, Sema, or Exen with Aβ40/Aβ42 abolished this effect. Meanwhile, the addition of GLP-1(7-37) increased the cytotoxicity of Aβ40 (Figure 7B) but did not affect the cytotoxic effect of Aβ42 (Figure 7C).

The most pronounced increase in the viability of SH-SY5Y cells was observed for Lira, Sema, and Exen upon treatment of the cells with Aβ42 (Figure 7C). Similarly, Sema reversed the effect of Aβ(25-35) on SH-SY5Y cells after their pretreatment with the latter [53]. Pretreatment of neuronal cells with Lira or Exen also protected them from Aβ(25-35) and Aβ42, respectively [30,37,81].

The protective effect of Exen on the Aβ-treated neuroblastoma cells was consistent with the results of the Aβ fibrillation experiments (Figure 4 and Figure 5) and rescuing memory deficits in AD mice [40]. However, such benefits have not been replicated in clinical trials [43]. Despite its higher affinity for monomeric Aβ, Lira exhibited a similar set of the properties, except for the lack of significant effect on Aβ40 fibrillation (Table 4). By contrast, both Sema and GLP-1(7-37) showed conflicting results in the Aβ fibrillation and Aβ cytotoxicity tests (Table 4), which may reflect differences in the cytotoxic properties of the multimeric forms of Aβ formed in their presence. In the case of Sema, the rapid fibrillation of Aβ40 (Figure 4B) may prevent the accumulation of the more cytotoxic Aβ40 oligomers [9,79,80]. The excess of the latter in the case of GLP-1(7-37) appeared to favor its cytotoxicity, despite the suppression of Aβ40 fibrillation in its presence (Table 4).

## 3. Materials and Methods

### 3.1. Materials

Lira (Victoza, 6 mg/mL) and Sema (Ozempic, 1.34 mg/mL) were bought from Novo Nordisk (Bagsværd, Denmark). Exen (Byetta, 250 µg/mL) was obtained from Astra Zeneca (Cambridge, UK) and Acmec Biochemical Technology Co., Ltd. (Shanghai, China). GLP-1(7-37) was purchased from Merck KGaA (Darmstadt, Germany), cat. #G9416, and Aladdin (Riverside, CA, USA), cat. #G-118964. Lira, Sema, and Exen were dialyzed three times against 1000-fold excess of deionized water and then dialyzed twice against 50 mM Tris-HCl, 280 mM NaCl, 9.8 mM KCl, 5 mM CaCl_2_, 2 mM MgCl_2_, pH 7.4 (buffer A) for all experiments except for the BLI and SPR experiments.

Human Aβ40/Aβ42 was expressed in *E. coli* and purified as described earlier [82]. Briefly, the chimera of Aβ with ubiquitin was purified using Ni-NTA affinity chromatography and cleaved with Usp2-cc protease (prepared mainly as described in ref. [83]), followed by purification using Ni-NTA and C18 columns. The quality of the Aβ samples was controlled by SDS-PAGE and electrospray ionization mass spectrometry.

Protein concentrations were measured spectrophotometrically using molar extinction coefficients at 280 nm calculated according to ref. [84]: 6990 M^−1^cm^−1^ for Sema and Lira, 5500 M^−1^cm^−1^ for Exen, and 1490 M^−1^cm^−1^ for Aβ40/Aβ42 at pH 7.4–8.0.

Ethylenediaminetetraacetic acid (EDTA), magnesium chloride, thioflavin T (ThT), ethanolamine, and polyethylene glycol sorbitan monolaurate (TWEEN^®^ 20) were obtained from Merck KGaA (Darmstadt, Germany). 2-mercaptoethanol (2-ME) was obtained from Amresco^®^ LLC (Vienna, Austria). Urea, imidazole, sodium hydroxide, sodium dodecyl sulfate (SDS), and glycerol were purchased from PanReac AppliChem (Barcelona, Spain). Calcium/magnesium chloride were obtained from Honeywell Fluka (Charlotte, NC, USA). AbiFlow 100 Ni-NTA agarose was obtained from Abisense (Sirius, Russia). Hydrochloric acid was obtained from Sigma Tec LLC (Khimki, Russia). Ultra-grade Tris, HEPES, sodium chloride, and dimethyl sulfoxide (DMSO) were obtained from Helicon (Moscow, Russia). Trifluoroacetic acid (TFA) was purchased from Fisher Scientific Inc. (Waltham, MA, USA). Potassium chloride, Coomassie Brilliant Blue R-250, 3-(4,5-dimethylthiazol-2-yl)-2,5-diphenyltetrazolium bromide (MTT), and sodium azide were obtained from Dia-M (Moscow, Russia). Acetic acid and ammonium hydroxide were obtained from Chimmed (Moscow, Russia) and Component-reaktiv (Moscow, Russia), respectively. Dulbecco’s Modified Eagle Medium (DMEM), fetal bovine serum (FBS), and penicillin-streptomycin-glutamine were obtained from Gibco (New York, NY, USA). Ampicillin was bought from NeoFroxx (Einhausen, Germany). F12 was obtained from PanEco (Moscow, Russia). The stock solution of ThT (0.6 mg/mL) was prepared in deionized water. The ThT concentration was measured spectrophotometrically using the molar extinction coefficient at 412 nm of 36,000 M^−1^cm^−1^ [85].

Neuroblastoma SH-SY5Y cells were obtained from Prof. Valery P. Zinchenko (Institute of Cell Biophysics of the RAS, Pushchino, Russia).

### 3.2. BLI Measurements

GLP-1RAs were dialyzed three times against 1000-fold excess of deionized water and then dialyzed twice against 20 mM Tris-HCl, 140 mM NaCl, 4.9 mM KCl, 2.5 mM CaCl_2_, 1 mM MgCl_2_, pH 7.4 buffer for Exen and Lira or 20 mM HEPES-KOH, 140 mM NaCl, 4.9 mM KCl, 2.5 mM CaCl_2_, 1 mM MgCl_2_, pH 7.4 for Sema. GLP-1(7-37) was dissolved in the last buffer. The Aβ samples were pretreated by TFA and dissolved in DMSO (2 mg/mL) as described in ref. [58], and stored at −20 °C.

The affinity of Aβ40/Aβ42 (ligand) for Exen (4–15 μM), Lira (5–20 μM), Sema (17–38 μM), or GLP-1(7-37) (25–50 μM) (analyte) at 25 °C was measured by BLI using a ForteBio Octet^®^ QKe System (Fremont, CA, USA) in 96-well microplates with shaking at 1000 rpm. Aβ40/Aβ42 (0.05 mg/mL in 10 mM sodium acetate, pH 4.5 buffer) was immobilized on five amino-reactive biosensors while one reference sensor was loaded with 1 M ethanolamine solution through 1-ethyl-3-[3-dimethylaminopropyl]carbodiimide hydrochloride/N-hydroxysulfosuccinimide (EDAC/sulfo-NHS) reaction until the Aβ40/Aβ42 loading level of 3.5 nm was reached. The rest of the activated amine groups on the biosensors were blocked by 1 M ethanolamine solution. The non-covalently bound Aβ40/Aβ42 molecules were washed off with 0.5% SDS and then with assay buffer (20 mM HEPES-KOH/Tris-HCl, 140 mM NaCl, 4.9 mM KCl, 2.5 mM CaCl_2_, 1 mM MgCl_2_, pH 7.4). The loading level after washing was 1.5 nm. The baseline collection time was 300 s, association with an analyte in the assay buffer was recorded for 600 s, and the dissociation phase was 600 s or 1200 s. The ligand was regenerated by triple immersion in 0.1% SDS water solution for 5 sec, followed by a 30 s rinsing with assay buffer. The BLI signal was corrected for baseline drift and non-specific binding by subtraction of the signal from the reference sensor, and fit to the *single binding site* model (A, analyte; L, ligand):(1)A+Lka⇄kdAL  KD
or the heterogeneous ligand scheme:(2)A+L1ka1⇄kd1AL1 A+L2ka2⇄kd2AL2KD1KD2
where *k_a_* and *k_d_* are the kinetic association and dissociation constants, respectively, and *K_D_* is the equilibrium dissociation constant. The constants were evaluated for each analyte concentration using ForteBio Data Analysis software v.12.0 (Fremont, CA, USA); standard deviations are indicated.

### 3.3. SPR Measurements

SPR studies of Exen, Sema, and Lira interactions with monomeric Aβ40/Aβ42 were performed at 25 °C using a Bio-Rad ProteOn™ XPR36 instrument (Bio-Rad Laboratories, Inc., Hercules, CA, USA) mainly according to ref. [58]. Lira and Sema were exhaustively dialyzed against 10 mM sodium phosphate, 50 mM NaCl, pH 7.0 buffer. Exen was exhaustively dialyzed against buffer A. The concentrations of the stock solutions were 70–243 µM for Sema, 1.4–2.0 mM for Lira, and 27 µM for Exen. The Aβ samples were pretreated by TFA and dissolved in DMSO (2 mg/mL) as described in ref. [58], and stored at −20 °C.

Ligand (50 µg/mL Aβ40/Aβ42) was immobilized on the surface of a ProteOn GLH sensor chip (Bio-Rad Laboratories, Inc., Hercules, CA, USA) by amine coupling using EDAC/sulfo-NHS, with subsequent blocking of the remaining activated amine groups on the chip surface by 1 M ethanolamine solution. The noncovalently bound Aβ40/Aβ42 molecules were washed off the chip surface with 0.5% SDS water solution. Analyte (0.0625–2 µM Sema, 1–8 µM Lira, and 0.5–6 µM Exen) in the running buffer (10 mM HEPES-NaOH, 150 mM NaCl, 0.05% Tween 20, pH 7.4) was passed over the sensor at a rate of 30 µL/min for 300 s (association phase), followed by flushing the chip with the running buffer for 900 s (dissociation phase). The ligand was regenerated by passage of 10 mM glycine, pH 3.3 buffer for 50 s. The kinetic SPR data were corrected for baseline drift and non-specific binding, and described using a *heterogeneous ligand* model (2). The *k_a_*, *k_d_*, and *K_D_* values were estimated using Bio-Rad ProteOn Manager™ v.3.1 software (Bio-Rad Laboratories, Inc., Hercules, CA, USA). The estimates were performed for each dataset globally, followed by their averaging; standard deviations are indicated.

### 3.4. Dynamic Light Scattering Measurements

DLS measurements were carried out using a Zetasizer Nano ZS system (Malvern Instruments Ltd., Malvern, UK). The backscattered light from a 4 mW He-Ne laser at 632.8 nm was collected at an angle of 173°. Lira (6–105 µM), Exen (15–234 µM), Sema (12–47 µM), and GLP-1(7-37) (5–83 µM) solutions in 25 mM Tris-HCl, 140 mM NaCl, 4.9 mM KCl, 2.5 mM CaCl_2_, 1 mM MgCl_2_, pH 7.4 buffer were incubated at 25 °C for 3 min. The acquisition time for a single autocorrelation function was 100 s. The resulting autocorrelation functions were averaged values from three measurements. The volume-weighted size distributions were calculated using the following parameters for the buffer: the refractive index of 1.334 was measured using the RL3 refractometer (PZO, Warszawa, Poland); the viscosity value η = 0.95 mPa∙s was measured using a micro-rheology method with a water suspension of standard latex nanoparticles (NIST 3060A, Thermo Fisher Scientific, Waltham, MA, USA). The molecular mass and its standard deviation corresponding to the volume-weighted hydrodynamic radius *MW_Rh_* distribution was calculated in approximation of a globular protein according to the equations from ref. [86]. The degree of multimerization was calculated as the *MW_Rh_/MW_m_* ratio, where *MW_m_* is the molecular mass of the monomeric GLP-1RAs calculated from their molecular structure.

### 3.5. Structural Modeling

The unrelaxed structure of Sema was built in PyMOL v.2.0 software (Schrödinger, Inc., New York, NY, USA) based on the structure of Sema- and taspoglutide-bound GLP-1 receptor in complex with Gs protein (PDB ID: 7KI0, EM, chain E) by combination with a linker and C18 di-acid chain (Figure 1B). The structures of the linker and C18 di-acid chain were built using ChemDraw v.22 (Boston, MA, USA) and minimized in the MM2 field (ChemDraw v.22).

The tertiary structures of Aβ40 (PDB ID 2LFM, NMR, model 1), Aβ40 fibril (PDB ID 2LMN, NMR, model 1), Exen (PDB ID 1JRJ, NMR, chain A, model 1), and GLP-1(7-37) (PDB ID 3IOL, X-ray, chain B) were taken from PDB (www.rcsb.org, accessed on 1 March 2025 [87]). The models of the tertiary structures of Aβ40/protofibril complexes with Exen or GLP-1(7-37) were built using the ClusPro docking server [78]. The resulting complexes were visualized and analyzed using PyMOL v.2 (https://pymol.org (accessed on 1 March 2025)). The contact residues in the docking models were calculated using a PyMOL script. The numbering of the contact residues is according to the PDB entries.

### 3.6. ThT Fluorescence Assay

ThT fluorescence emission measurements were carried out mainly as described in ref. [82] using a BioTek Synergy H1 microplate reader (Agilent Technologies, Inc., Santa Clara, CA, USA) with an emission wavelength of 485 nm and excitation at 440 nm. The Aβ40 sample was prepared as described in ref. [82] with some modifications (~5 mM NaOH at pH 11.8, 0.5 mg/mL). Aβ40 (20 µM) was incubated at 30 °C in the absence/presence of 10 µM Sema, Lira, Exen, or GLP-1(7-37). The control curves (without Aβ40; Appendix A) were subtracted from the corresponding kinetic curves of Aβ40 samples with/without GLP-1RAs. Each measurement was performed in 2–10 repetitions. The mean fluorescence signal values for each experimental sample were normalized to the average fluorescence signal corresponding to the saturation phase of Aβ40 fibril formation without additives. Data are presented as mean ± standard deviation.

### 3.7. Transmission Electron Microscopy

A copper grid (300-mesh) coated with a 0.2% formvar film was placed on a 10 µL drop of the sample. After incubating the sample (following the ThT fluorescence assay) for 15 min to allow adsorption, the grid was stained with a 1% (*w/v*) aqueous solution of uranyl acetate for 2 min. Excess stain was removed using filter paper, and the grid was rinsed in deionized water for 1 min. The samples were analyzed using a JEM-1400Plus (HC) transmission electron microscope (JEOL, Ltd., Tokyo, Japan) at an accelerating voltage of 80 keV.

### 3.8. Cell Viability Assay

Human neuroblastoma SH-SY5Y cells were cultured in DMEM-F12 medium supplemented with 1% penicillin-streptomycin-glutamine and 10% fetal bovine serum at 37 °C for 24 h in a humidified atmosphere with 5% CO_2_. Upon reaching 80% confluence, the cells were harvested and seeded into 96-well plates at a density of 15 × 10^5^ cells per well in serum-free DMEM/F12 + PSG medium.

The Aβ40/Aβ42 samples were dissolved in fresh 1% NH_4_OH at a concentration of 0.5 mg/mL, followed by freeze-drying. The dried Aβ40/Aβ42 samples were dissolved in serum-free DMEM medium at a concentration of 40–50 µM (0.17–0.23 mg/mL), followed by mixing with the GLP-1RA (Sema, Lira, Exen or GLP-1) stock solutions in buffer A and the same medium to a final concentration of both components of 20 µM.

Freshly prepared Aβ40/Aβ42, GLP-1RAs, or their mixtures were added (100 µL per well) to the cultures 24 h after seeding to a final concentration of both components of 10 µM. The final volume of medium in the well was 200 µL. The MTT assay, designed to assess cellular metabolic activity, was performed after incubation of the cells for 48 h. MTT (0.005 mg/mL) was added and the cells were incubated for 3 h, followed by solubilization of the cells using DMSO. The absorbance at 550 nm was measured using an BioTek Synergy H1 microplate reader (Agilent Technologies, Inc., Santa Clara, CA, USA). The resulting values were normalized relative to the control group of the untreated cells (100%). Data are presented as mean ± standard deviation (*n* = 5–10).

## 4. Conclusions

The risk of AD development in patients with diabetes increases by approximately 65% [8] since these diseases share some pathological features [12]. Therefore, antidiabetic drugs, including GLP-1RAs, are now being repurposed for the treatment of AD [14]. Although both animal studies and clinical trials have reported beneficial effects of GLP-1RAs on the course of AD [29,36,85], the molecular mechanisms underlying these effects remain poorly understood. Here, we demonstrated direct interactions between GLP-1RAs such as GLP-1(7-37), Lira, Sema, and Exen with the monomeric forms of Aβ40 and Aβ42 under in vitro conditions mimicking physiological conditions. Comparison of the *K_D_* estimates for Aβ-Sema/Lira complexes with peak plasma concentrations of Sema/Lira indicated the potential physiological significance of these interactions. This suggestion is supported by the marked effect of Sema on Aβ40 fibrillation in vitro and the effect of Sema/Lyra on Aβ-induced cytotoxicity toward SH-SY5Y cells. Similarly, Exen and GLP-1(7-37) affect Aβ40 fibrillation and cytotoxicity of Aβ toward SH-SY5Y cells. Notably, these effects largely depend on the specific GLP-1RA and do not necessarily correlate with the results of animal and clinical studies (Table 4). The latter also depends on the ability of the GLP-1RA to cross the BBB, which is only possessed by Exen, Lira, and GLP-1(7-37), but not by Sema.

Our findings indicate that, despite certain structural similarities, individual GLP-1RAs exhibit distinct behaviors in vitro in terms of their affinity for Aβ, their influence on Aβ fibril formation, and their modulation of Aβ-associated cytotoxicity (Table 4). Further clinical trials of GLP-1-based drugs are needed to rule out the possibility of neuronal damage that does not necessarily lead to progression of AD. Our findings not only suggest a new mechanism for the influence of GLP-1RAs on Aβ metabolism in vivo but also provide a basis for the development of GLP-1RA drugs with more pronounced anti-AD effects.

## Figures and Tables

**Figure 1 ijms-26-04095-f001:**
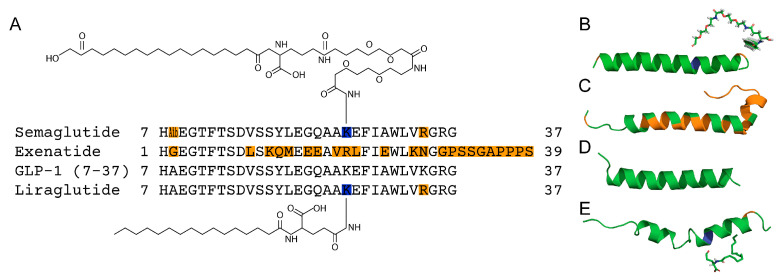
The alignment of amino acid sequences (panel **A**) and model structures of GLP-1RAs (**B**–**E**): Sema (**B**: based on PDB entry 7KI0, EM, chain **E**), Exen (PDB ID 1JRJ, NMR, chain **A**, model 1), GLP-1(7-37) (**D**: PDB ID 3IOL, X-ray, chain **B**), and Lira (**E**: PDB ID 4APD, NMR, chains **A**, **B**, model 1). The amino acid residues that differ from those in GLP-1 are marked in orange (Aib, 2-aminoisobutyric acid). The lysine residues of Sema and Lira modified by the linkers with fatty acids are highlighted in blue. The numbering of the residues is according to the PDB entries.

**Figure 2 ijms-26-04095-f002:**
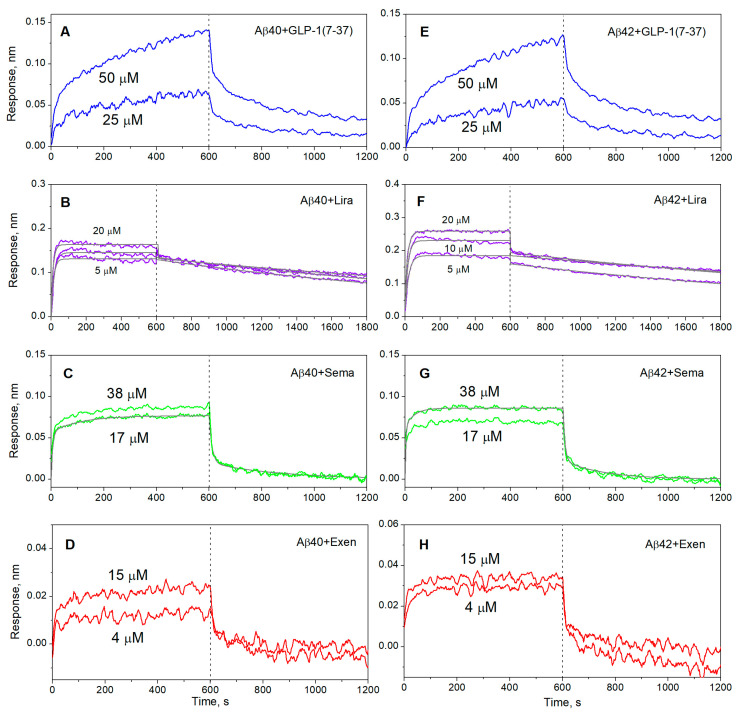
Kinetics of GLP-1(7-37) (blue), Exen (red), Lira (violet), and Sema (green) interactions with monomeric Aβ40 (panels **A**–**D**) or Aβ42 (**E**–**H**) immobilized on the sensor surface by amine coupling, monitored using BLI at 25 °C (20 mM HEPES-KOH/Tris-HCl, 140 mM NaCl, 4.9 mM KCl, 2.5 mM CaCl_2_, 1 mM MgCl_2_, pH 7.4). The analyte concentrations are indicated near the sensograms. The black curves are theoretical, calculated according to the single binding site scheme (1) or heterogeneous ligand model (2) (see Table 1 for the fitting parameters).

**Figure 3 ijms-26-04095-f003:**
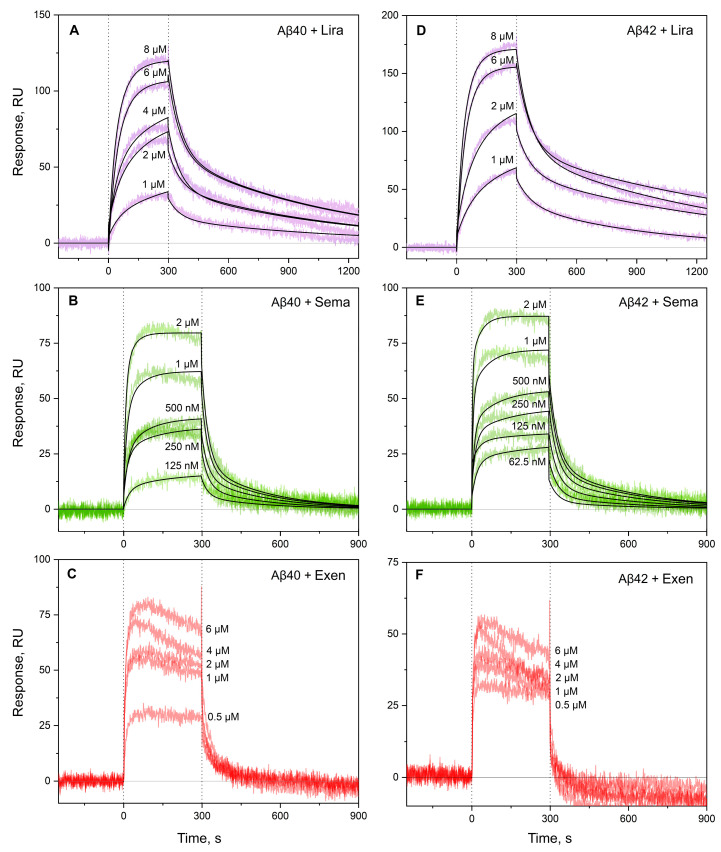
Kinetics of interactions between monomeric Aβ40 (panels **A**–**C**) or Aβ42 (**D**–**F**) and Lira (violet), Sema (green), or Exen (red) at 25 °C, monitored using SPR (10 mM HEPES-NaOH, 150 mM NaCl, 0.05% Tween 20, pH 7.4). The analyte concentrations are indicated near the sensograms. The black curves are theoretical, calculated according to the heterogeneous ligand model (2) (see Table 2 for the fitting parameters).

**Figure 4 ijms-26-04095-f004:**
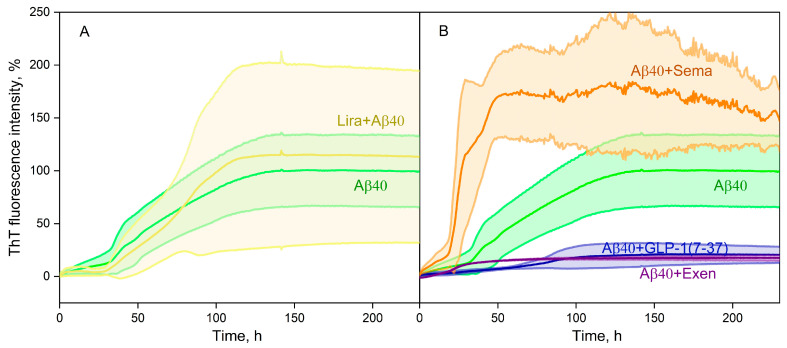
Kinetics of fluorescence intensity at 485 nm with 10 μM ThT added to 20 µM Aβ40 in the absence or in the presence of 10 µM Lira (**A**), Sema, Exen, or GLP-1(7-37) (**B**) at 30 °C (25 mM Tris-HCl, 140 mM NaCl, 4.9 mM KCl, 2.5 mM CaCl_2_, 1 mM MgCl_2_, 0.05% NaN_3_, pH 7.4). Standard deviations of the fluorescence signals are indicated. Excitation wavelength of 440 nm.

**Figure 5 ijms-26-04095-f005:**
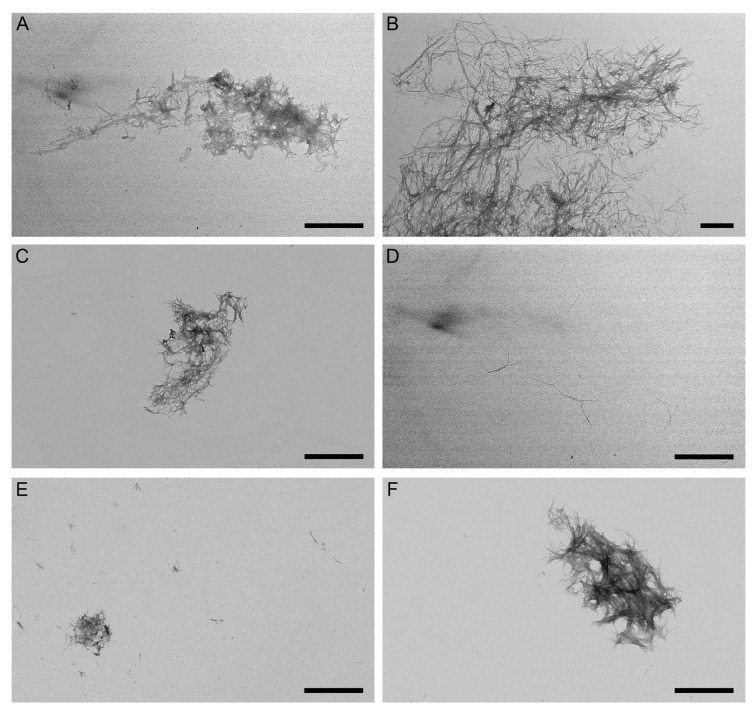
Negative-staining TEM images of Aβ40 fibers grown in the course of the ThT fluorescence assay shown in Figure 4 in the absence (panel **A**) or in the presence of 10 μM Lira (**B**), 10 μM Sema (**C**), 10 μM Exen (**D**), and GLP-1(7-37) (**E**,**F**). The scale bars represent 1 μm.

**Figure 6 ijms-26-04095-f006:**
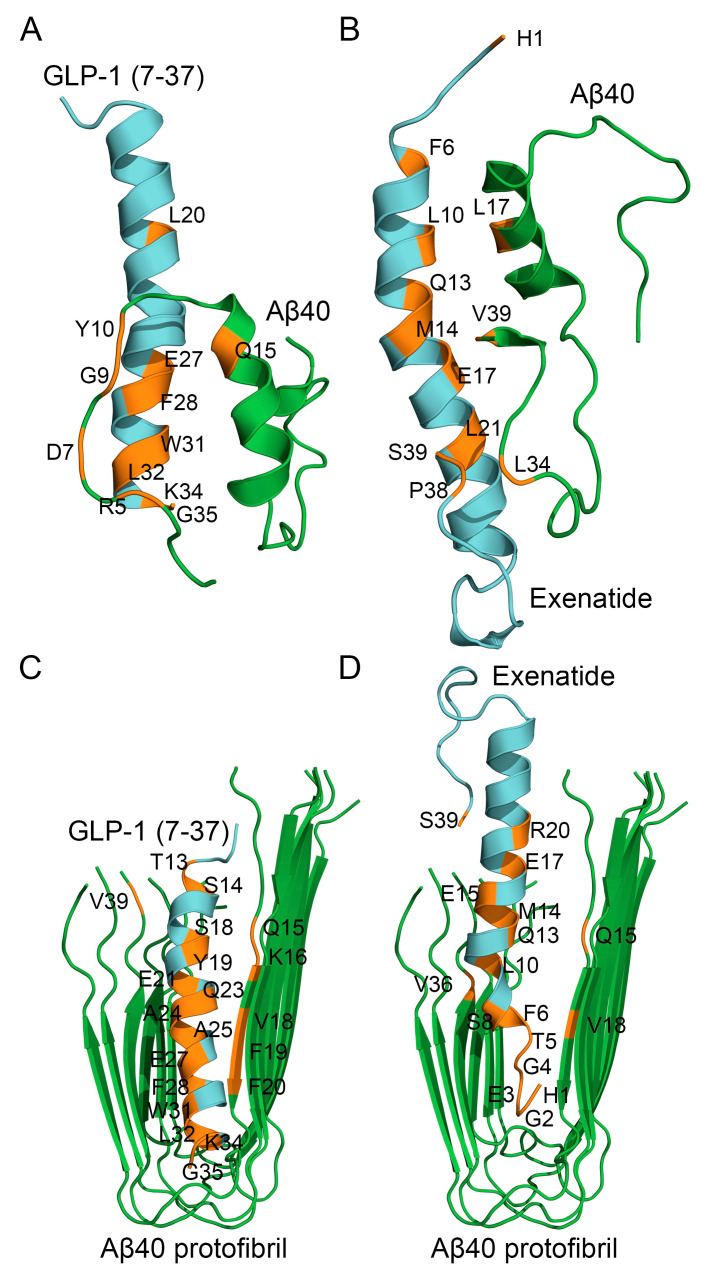
The models of tertiary structures of the complexes between monomeric Aβ40 (taken from PDB entry 2LFM, model 1) or Aβ40 protofibril (PDB entry 2LMN, model 1) (shown in green) and GLP-1(7-37) (PDB entry 3IOL, chain B) (panels **A**,**C**) or Exen (PDB entry 1JRJ, chain A) (**B**,**D**) (colored cyan) built using the ClusPro docking server. The contact residues are shown in orange. The numbering of the residues is according to the PDB entries.

**Figure 7 ijms-26-04095-f007:**
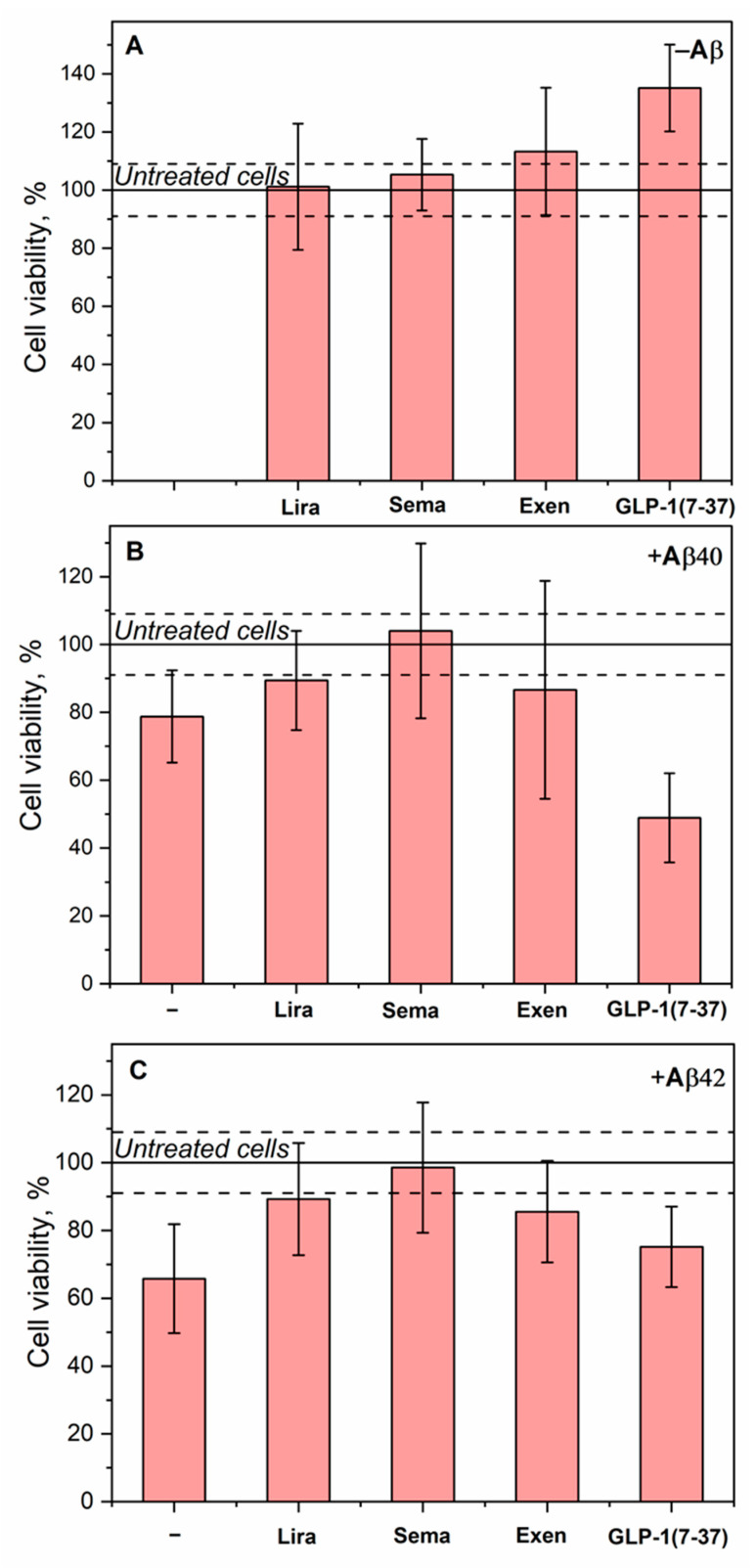
Effect of 10 µM Aβ40/Aβ42 and/or 10 µM Lira/Sema/Exen/GLP-1(7-37) on the viability of SH-SY5Y cells assessed by the MTT assay (serum-free medium, 5% CO_2_, 37 °C, incubation for 48 h). (**A**) Effect of GLP-1RAs on cell viability. (**B**) Effect of Aβ40 and GLP-1RAs in the presence of Aβ40 on cell viability. (**C**) Effect of Aβ42 and GLP-1RAs in the presence of Aβ42 on cell viability. Mean values and standard deviations are indicated. Solid and dashed lines indicate mean values and standard deviations for untreated cells, respectively.

**Table 1 ijms-26-04095-t001:** Parameters of the interactions between monomeric Aβ40/Aβ42 and GLP-1RAs at 25 °C, estimated from the BLI data shown in Figure 2 using either the single binding site model (1) or heterogeneous ligand scheme (2).

	[Lira], µM	*k_a_*, M^−1^s^−1^	*k_d_*, s^−1^	***K_D_*, M**	*k_a_*, M^−1^s^−1^	*k_d_*, s^−1^	***K_D_*, M**
Lira		Aβ40			Aβ42		
	20	(8.4 ± 2.8) × 10^3^	(9.0 ± 0.4) × 10^−4^	**(1.1 ± 0.4) × 10^−7^**	(5.7 ± 1.1) × 10^3^	(6.0 ± 0.2) × 10^−4^	**(1.1 ± 0.2) × 10^−7^**
	10	(7.3 ± 0.4) × 10^3^	(3.46 ± 0.07) × 10^−4^	**(4.8** **± 0.3) ×** **10^−8^**	(8.0 ± 0.7) × 10^3^	(4.82 ± 0.12) × 10^−4^	**(6.0 ± 0.2) × 10^−8^**
	5	(1.34 ± 0.09) × 10^4^	(5.56 ± 0.11) × 10^−4^	**(** **4.2 ± 0.3) × 10^−8^**	(1.21 ± 0.08) × 10^4^	(5.40 ± 0.12) × 10^−4^	**(** **4.5 ± 0.3) × 10^−8^**
	[Sema], µM	*k_a_*_1_, M^−1^s^−1^	*k_d_*_1_, s^−1^	***K_D_*_1_, M**	*k_a_*_2_, M^−1^s^−1^	*k_d_*_2_, s^−1^	** *K_D_* ** **_2_, M**
Sema		Aβ40					
	17	310 ± 52	(3.7 ± 0.2) × 10^−3^	**(1.2 ± 0.2) × 10^−5^**	(5.7 ± 1.1) × 10^3^	(9.1 ± 0.6) × 10^−2^	**(1.6 ± 0.3) × 10^−5^**
		Aβ42					
	38	582 ± 104	(6.4 ± 0.6) × 10^−3^	**(1.1 ± 0.2) × 10^−5^**	(6.1 ± 2.7) × 10^3^	(1.34 ± 0.15) × 10^−1^	**(2.2 ± 1.0) × 10^−5^**

**Table 2 ijms-26-04095-t002:** Parameters of the interactions between monomeric Aβ40/Aβ42 and GLP-1RAs at 25 °C, estimated from the SPR data shown in Figure 3 using the heterogeneous ligand model (2).

	[GLP-1RA], μM	*k_a_*_1_, M^−1^s^−1^	*k_d_*_1_, s^−1^	***K_D_*_1_, M**	*k_a_*_2_, M^−1^s^−1^	*k_d_*_2_, s^−1^	** *K_D_* ** **_2_, M**
Aβ40							
Sema	0.06–2	(9.6 ± 2.4) × 10^3^	(3.2 ± 0.9) × 10^−3^	**(3.4 ± 0.5) × 10^−7^**	(3.90 ± 1.12) × 10^4^	(4.32 ± 0.12) × 10^−2^	**(1.2 ± 0.4) × 10^−6^**
Lira	1–8	(1.44 ± 0.05) × 10^3^	(1.19 ± 0.13) × 10^−3^	**(9.5 ± 0.7) × 10^−7^**	(1.9 ± 0.3) × 10^3^	(1.70 ± 0.10) × 10^−2^	**(9.1 ± 1.6) × 10^−6^**
Aβ42							
Sema	0.06–2	(1.26 ± 0.11) × 10^4^	(4.1 ± 1.4) × 10^−3^	**(3.4 ± 1.4) × 10^−7^**	(1.34 ± 0.18) × 10^5^	(3.8 ± 0.7) × 10^−2^	**(3.0 ± 0.9) × 10^−7^**
Lira	1–8	(2.24 ± 0.18) × 10^3^	(1.14 ± 0.10) × 10^−3^	**(5.2 ± 0.8) × 10^−7^**	(2.9 ± 0.04) × 10^3^	(1.64 ± 0.14) × 10^−2^	**(5.6 ± 0.3) × 10^−6^**

**Table 3 ijms-26-04095-t003:** Concentration dependence of hydrodynamic radius (*R_h_*), molecular mass (*MW_Rh_*), and degree of multimerization (*MW_Rh_*/*MW_m_*) for GLP-1RAs at 25 °C, determined by DLS (25 mM Tris-HCl, 140 mM NaCl, 4.9 mM KCl, 2.5 mM CaCl_2_, 1 mM MgCl_2_, pH 7.4).

GLP-1RA	[GLP-1RA], µM	*R_h_*, nm	*MW_Rh_*, kDa	*MW_Rh_*/*MW_m_*
GLP-1(7-37)	5–83	>92	>7 × 10^5^	>210
Lira	105	3.08 ± 0.15	54.7 ± 7.8	15.6 ± 2.2
	52	3.13 ± 0.05	57.1 ± 2.6	16.3 ± 0.7
	13	2.25 ± 0.12	22.7 ± 6.1	6.5 ± 1.7
	6	2.45 ± 0.16	28.8 ± 3.6	8.2 ± 1.0
Exen	234	2.20 ± 0.01	24.58 ± 0.02	5.88 ± 0.04
	115	2.27 ± 0.16	26.7 ± 5.7	6.4 ± 1.4
	29	1.53 ± 0.06	8.8 ± 0.9	2.1 ± 0.2
	15	1.39 ± 0.07	6.7 ± 0.9	1.6 ± 0.2
Sema	47	1.22 ± 0.04	4.1 ± 0.4	1.2 ± 0.1
	12	1.42 ± 0.15	6.2 ± 2.1	1.9 ± 0.6

**Table 4 ijms-26-04095-t004:** Summary of the properties of the GLP1-RAs obtained in this work and described in the literature. The counteracting effects are highlighted in bold.

	Minimal *K_D_* for Aβ Binding According to BLI	Effect on Aβ40 Fibrillation (Figure 4 and Figure 5)	Effect on Aβ Cytotoxicity to SH-SY5Y Cells (Figure 7)	Ability to Cross the BBB	AD Animal Data	Clinical Data, AD
Lira	4.2 × 10^−8^ M	No effect	Protection	+ [47]	Prevents memory loss, reduces Aβ amyloid deposits [47,48,49]	No effect [50]
Sema	1.1 × 10^−5^ M	**Stimulation**	**Protection**	− [74]	Positive effects on cognitive function, reduction of Aβ amyloid deposits [55]	Phase 3 clinical trials (NCT04777396 and NCT04777409)
Exen	~(0.4–1.5) × 10^−5^ M	Inhibition	Protection	+ [73]	Positive effects on learning and memory ability, reduces Aβ deposition [38,39,40,41]	No effect [43]
GLP-1(7-37)	~(2.5–5.0) × 10^−5^ M	**Inhibition**	**Increases Aβ40 cytotoxicity**	+ [24]	Positive effects on learning and memory [29]	No data

## Data Availability

Data are contained within the article and Appendix A.

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
