# Peer review of "Interaction Between Glucagon-like Peptide 1 and Its Analogs with Amyloid-β Peptide Affects Its Fibrillation and Cytotoxicity"

_ijms, 2025, doi:10.3390/ijms26094095_

Round 1

Reviewer 1 Report

Comments and Suggestions for Authors

The research results included in the manuscript are a continuation of the scientific trend showing that some antidiabetic drugs, including glucagon-like peptide 1 receptor agonists (GLP-1RA), exert therapeutic effects in Alzheimer's disease (AD) by modulating the metabolism of β-amyloid peptide (Aβ). The authors presented the interactions and effects of glucagon-like peptide 1 and its analogues on the activity of β-amyloid peptide 2 (i.e., its fibrillation and cytotoxicity). The work was prepared meticulously and accurately in every part of the manuscript. The introduction was supported by numerous references and accurately reflects the idea of ​​the research. The results were presented honestly and clearly, and the whole was summarized in a concise conclusion summarizing the research. A supplement was also prepared. The language side should be corrected, e.g. in words of Latin origin, italics are used.

Author Response

Comments: The research results included in the manuscript are a continuation of the scientific trend showing that some antidiabetic drugs, including glucagon-like peptide 1 receptor agonists (GLP-1RA), exert therapeutic effects in Alzheimer's disease (AD) by modulating the metabolism of β-amyloid peptide (Aβ). The authors presented the interactions and effects of glucagon-like peptide 1 and its analogues on the activity of β-amyloid peptide 2 (i.e., its fibrillation and cytotoxicity). The work was prepared meticulously and accurately in every part of the manuscript. The introduction was supported by numerous references and accurately reflects the idea of ​​the research. The results were presented honestly and clearly, and the whole was summarized in a concise conclusion summarizing the research. A supplement was also prepared. The language side should be corrected, e.g. in words of Latin origin, italics are used.

Answer: Thank you for your assistance with the manuscript and for your positive evaluation of our work. The language has now been edited.

Reviewer 2 Report

Comments and Suggestions for Authors

Please confirm if is it really 416 million people worldwide with Alzheimer’s disease dementia. I guess it is around 55 million. Please provide your references or change the statement accordingly. Further, it was mentioned “22% of all people aged 50 and older”. Please also confirm this percentage concerning what?

Are aducanumab and lecanemab both FDA-approved? What are their mechanism of action and associated problems that necessities further investigation to combat AD?

Wasn’t there any possibility of conducting an in vivo animal study?

The number of keywords can be reduced.

Author Response

Comments 1: Please confirm if is it really 416 million people worldwide with Alzheimer’s disease dementia. I guess it is around 55 million. Please provide your references or change the statement accordingly. Further, it was mentioned “22% of all people aged 50 and older”. Please also confirm this percentage concerning what?

Answer 1: This quantity includes people with preclinical stage Alzheimer's disease (AD) (315 million), prodromal AD (69 million) and AD dementia (32 million). The total of 416 million represents 22% of the global population over the age of 50. The phrasing in the text has been updated for clarity and informativeness.

Comments 2: Are aducanumab and lecanemab both FDA-approved? What are their mechanism of action and associated problems that necessities further investigation to combat AD?

Answer 2: Aducanumab has received accelerated approval from the FDA in 2021 (https://www.fda.gov/drugs/postmarket-drug-safety-information-patients-and-providers/aducanumab-marketed-aduhelm-information). Later lecanemab and donanemab have received traditional approval (https://www.fda.gov/news-events/press-announcements/fda-converts-novel-alzheimers-disease-treatment-traditional-approval; https://www.fda.gov/drugs/news-events-human-drugs/fda-approves-treatment-adults-alzheimers-disease). These drugs are designed to reduce amyloid deposits, but their use carries a risk of brain edema and hemorrhage. We have provided additional details on these drugs in the text. We agree that further research is essential to identify the most effective therapeutic strategies while minimizing side effects. For example, studies have shown that certain side effects occur more frequently in individuals with the apolipoprotein E4 (ApoE4) allele (https://doi.org/10.1186/s41983-024-00845-5), which warrants further investigation.  

Comments 3: Wasn’t there any possibility of conducting an in vivo animal study?

Answer 3: Unfortunately, we currently do not have the opportunity to do so. However, we hope to address this issue in the future.

Comments 4: The number of keywords can be reduced.

Answer 4: Done

We appreciate the work you have done, which has helped improve the manuscript.